# Evaluating Completeness of Foodborne Outbreak Reporting in the United States, 1998–2019

**DOI:** 10.3390/ijerph19052898

**Published:** 2022-03-02

**Authors:** Yutong Zhang, Ryan B. Simpson, Lauren E. Sallade, Emily Sanchez, Kyle M. Monahan, Elena N. Naumova

**Affiliations:** 1Division of Nutrition Epidemiology and Data Science, Tufts University Friedman School of Nutrition Science and Policy, 150 Harrison Avenue, Boston, MA 02111, USA; ryan.simpson@tufts.edu (R.B.S.); lauren.sallade@gmail.com (L.E.S.); emily.sanchez@tufts.edu (E.S.); 2Gordon Institute, Tufts University School of Engineering, 200 Boston Avenue, Medford, MA 02155, USA; kyle.monahan@tufts.edu

**Keywords:** data completeness, electronic Foodborne Outbreak Reporting System (eFORS), foodborne outbreaks, National Outbreak Reporting System (NORS), precision public health, time series analyses

## Abstract

Public health agencies routinely collect time-referenced records to describe and compare foodborne outbreak characteristics. Few studies provide comprehensive metadata to inform researchers of data limitations prior to conducting statistical modeling. We described the completeness of 103 variables for 22,792 outbreaks publicly reported by the United States Centers for Disease Control and Prevention’s (US CDC’s) electronic Foodborne Outbreak Reporting System (eFORS) and National Outbreak Reporting System (NORS). We compared monthly trends of completeness during eFORS (1998–2008) and NORS (2009–2019) reporting periods using segmented time series analyses adjusted for seasonality. We quantified the overall, annual, and monthly completeness as the percentage of outbreaks with blank records per our study period, calendar year, and study month, respectively. We found that outbreaks of unknown genus (*n* = 7401), *Norovirus* (*n* = 6414), *Salmonella* (*n* = 2872), *Clostridium* (*n* = 944), and multiple genera (*n* = 779) accounted for 80.77% of all outbreaks. However, crude completeness ranged from 46.06% to 60.19% across the 103 variables assessed. Variables with the lowest crude completeness (ranging 3.32–6.98%) included pathogen, specimen etiological testing, and secondary transmission traceback information. Variables with low (<35%) average monthly completeness during eFORS increased by 0.33–0.40%/month after transitioning to NORS, most likely due to the expansion of surveillance capacity and coverage within the new reporting system. Examining completeness metrics in outbreak surveillance systems provides essential information on the availability of data for public reuse. These metadata offer important insights for public health statisticians and modelers to precisely monitor and track the geographic spread, event duration, and illness intensity of foodborne outbreaks.

## 1. Introduction

Worldwide, public health agencies routinely collect time-referenced records to monitor ~600 million foodborne or waterborne outbreaks occurring annually [1,2,3,4,5]. In the United States (US) alone, approximately 1 in 6 Americans suffer from a foodborne illness resulting in ~48 million cases, ~128,000 hospitalizations, and ~3000 deaths annually [6]. Nearly 90% of these illnesses and hospitalizations are caused by five pathogens, including *Salmonella*, *Toxoplasma*, *Staphylococcus aureus*, *Norovirus*, and *Campylobacter* [6]. In 2013, the US Department of Agriculture (USDA) Economic Research Service (ERS) estimated that the frequency and severity of foodborne illnesses culminate in ~$15.5 billion (USD 2013) of losses annually attributed to medical costs, productivity losses, and economic burden due to death [7]. A recent 2021 report suggests that these expenses have risen by 13% to an estimated ~$17.6 billion per year (USD 2018) [7]. Much of these costs have been associated with food recalls and outbreaks attributed to salmonellosis (~$4.14 billion), toxoplasmosis (~$3.74 billion), listeriosis (~$3.19 billion), *Norovirus* (~$2.57 billion), and campylobacteriosis (~$2.18 billion) [8]. The high economic and health burdens of foodborne illnesses have demanded extensive passive surveillance to effectively monitor, track, and contain outbreaks. The compilation of complete and comprehensive information on causal agents and contributing factors is a challenging task, and few countries have a well-integrated surveillance system for foodborne infections due to the challenges of managing data effectively, the absence of early disease detection, inadequate computing resources, lack of financial support, and staff shortages [9,10,11,12].

The US Centers for Disease Control and Prevention (CDC) define foodborne outbreaks as two or more illnesses occurring in a short period of time due to the consumption of a common food or water source [13]. National outbreak surveillance began in 1971 with the Waterborne Disease Outbreak Surveillance System (WBDOSS), followed by the Foodborne Disease Outbreak Surveillance System (FDOSS) in 1973 [14]. Both systems used paper records to conduct event-based surveillance (pFORS) until 1998 when transitioning to the electronic Foodborne Outbreak Reporting System (eFORS) [14]. In 2009, the CDC integrated eFORS with other outbreak reporting systems under a single National Outbreak Reporting System (NORS), which also expanded to monitor outbreaks associated with person-to-person, animal, environmental, and unknown modes of transmission [14]. NORS has permitted cross-agency integration of other databases, including OHHABS (the One Health Harmful Algal Bloom System), CaliciNet (national *Norovirus* surveillance network), PulseNet (local, state, and federal public health laboratory network), and NARMS (the National Antimicrobial Resistance Monitoring System) [15]. NORS has continuously enhanced data collection and reporting protocols while improving the functionality and usability of available data [14]. Health departments report outbreaks using *NORSDirect*, which automatically uploads and registers single or multi-location outbreaks [16]. The transition to the NORS electronic system integrates and streamlines outbreak surveillance, enhances state and local outbreak reporting, and provides key temporal and spatial information to accompany illness counts, symptoms, pathogen etiology, and food/drink sources [14]. Electronic health records permit greater public sharing and extended longevity and usability of historical data [17]. Yet, public health professionals fail to compile complete and comprehensive records due to limited time and resources for traceback investigations, the complexity of these investigations, and challenges linking illness exposure, symptoms, and healthcare site services [18,19,20]. Furthermore, as a passive surveillance system, state, and local health agencies are encouraged but not required to report outbreak records to NORS [14]. While electronic reporting improves the precision of temporal and spatial information to accompany outbreak characteristics [14], national reports indicate significant underreporting of outbreaks due to this passive design [21].

Studies utilizing NORS have commonly explored three categories of available data: epidemiologic, contaminant traceback, and food/environmental testing [22]. Epidemiologic studies have described differences in outbreak characteristics by the geographic distribution of illnesses, time of exposure and illness onset, location of exposure, incubation periods, or food sources associated with illness [22]. Traceback studies have compared pathways of infection across outbreaks, including primary and secondary transmission contacts and how pathogens enter the food supply and spread once contaminating foods [22]. Lastly, food and environmental testing studies have evaluated differences in the burden of outbreaks according to etiological information such as pathogen genera or subtypes [22]. Despite its utility in exploring foodborne outbreak characteristics, some studies have noted the incompleteness of NORS records [23,24,25]. Completeness reflects the usability of available data, and its patterns influence the credibility of statistical analyses [26]. Patterns of incomplete records may distort seasonal patterns of infections, inhibiting researchers’ ability to track shifts in seasonal peak timing or assess associations between illness incidence and environmental drivers of infection [27]. We expand on this research by modifying completeness metrics to evaluate the credibility and usability of event-based surveillance data.

In this study, we developed a framework to perform systematic screening of data completeness to serve as metadata for outbreak surveillance systems. We evaluated the completeness of publicly reported foodborne outbreak data in eFORS and NORS event-based electronic surveillance systems and explored how the implementation of NORS improved data completeness. We extracted, aligned, and merged 25 data tables containing 213 variables for 22,792 outbreaks publicly reported from 1 January 1998 through 31 December 2019. We compared the patterns of completeness for 103 variables before and after the transition from eFORS (1998–2008) to NORS (2009–2019) using segmented linear regression models adapted to time-referenced monthly values and controlled for outbreak seasonality. Our results provide the basis for a standardized metadata report to accompany publicly available surveillance system data downloads to assist data users in effectively utilizing reported electronic health records.

## 2. Methods

### 2.1. Data Source

On 4 March 2021, we requested and received integrated NORS data for all available foodborne outbreak records from 1 January 1998 through 31 December 2019 [28], where records between 1 January to 31 December 2008 were collected by eFORS. Data were unavailable for 2020 and 2021 due to a ~12–18-month delay in the public distribution of outbreak records. Extracted data included 213 variables in 25 data tables broadly categorized by general (109 variables in 5 tables), etiological (48 variables in 2 tables), and food-related (56 variables in 18 tables) outbreak characteristics. As an event-based surveillance system, NORS recorded outbreaks using identification numbers (CDCID) to permit alignment and merging of variables across data tables. 

NORS records related to general outbreak information (e.g., location, illness date, incubation time, and case information) categorized primary cases by health outcomes (e.g., hospitalization, ER visits, etc.), age group (groups vary between eFORS and NORS), gender (i.e., male, female, unknown), and case definition (e.g., confirmed, probable, estimated). NORS etiological information described clinical and environmental testing procedures, sampling techniques, and pathogen etiology. Food-related characteristics pertained to the suspected modes of infection transmission, point of contamination in the supply chain, and where the contaminated foods were prepared or consumed. Some tables described only small subsets of observations, such as school-related outbreaks (3 tables), ground beef information (1 table), or egg information (1 table). Given the variability of food ingredients per outbreak, NORS provided an implicated food identification number (FID) for merging 7 ingredient-related data tables.

### 2.2. Data Preparation

NORS reported 3 types of variables, including strings (dates, text answers, notes), binary choice (dichotomous—0 for absent, 1 for present), and multiple-choice responses. During data pre-processing, the CDC converted multiple-choice questions either to numerous binary-choice variables or repeated outbreak observations under the same CDCID for each categorical option selected. The latter introduced a repeated observation structure that duplicated outbreak records within our dataset and required more extensive data cleaning and transposition to create a uniformly structured dataset. For these multiple-choice questions, data completeness depended on the ratio of options selected to the total response options available. For example, incomplete “case information for signs or symptoms of illnesses” could not exceed the number of “signs or symptoms” available for reporting per outbreak. Similarly, ingredient-related variables depended on the number of ingredients involved in an outbreak. We define these variables whose completeness depended on the ratio of categorical options selected to all those available per outbreak as conditional variables. 

To generate a single dataset, we cleaned and merged the 213 variables in all 25 data tables across 22,792 outbreaks (Appendix A). During our cleaning process, we first excluded: 26 variables generated by CDC personnel to track the reporting of electronic records (e.g., data recorder ID, local report date, CDC report date, etc.);12 variables providing reporter contact information and optional comments written during reporting (e.g., recall comments, agency title, reporting site, etc.);9 variables providing clarification responses to specific questions asked only for specific outbreaks (e.g., clarification of supply chain stage of contamination, questions regarding antimicrobial resistance testing, etc.); and17 variables unavailable for the entire study period duration (e.g., illness attack rate, percentage of illnesses by age group, food contaminant infecting exposed persons, age percentage, etc.).

Next, we collapsed 60 variables relating to multiple-choice questions into 14 variables estimated as the count of multiple-choice options per question. After completing this process, the final dataset consisted of 103 variables (Figure 1). We calculated completeness as the ratio for which variable information was reported per outbreak. 

### 2.3. Crude Completeness Estimation

For each outbreak, we determined whether any of the 103 variables had a complete, partial, or absent record. We differentiated between incomplete records as no information available for a variable (e.g., blanks) and values of 0. For all but conditional variables, we created a dichotomous indicator defined as 1 if an outbreak had complete information for that variable and 0 if not. Dichotomous indicators were left blank if variables did not pertain to an outbreak, such as clarification questions asked only to a subset of outbreaks (i.e., the handling of beef food products for non-beef-associated outbreaks). Indicator blanks properly corrected completeness estimates for only those outbreaks eligible to report information on a given variable. For conditional variables, we estimated completeness as the ratio of reported categorical responses to the total responses available per outbreak. Dichotomous indicators contained 4 types of information: completely missing (0), partially missing (ranging 0–1), and non-missing records (1), and records where completeness information was not applicable for a given outbreak (blanks).

Comparing variables’ completeness between eFORS and NORS required additional data aggregation for some variables. We identified differences in age group definitions between eFORS (6 groups: <1, 1–4, 5–19, 20–49, ≥50, unknown years) and NORS (8 groups: <1, 1–4, 5–9, 10–19, 20–49, 50–74, 75+, unknown years). To standardize completeness estimates across surveillance periods, we calculated the average completeness across all age groups.

We measured crude completeness across outbreaks and variables. Crude outbreak completeness (*C_i_*) reflected the percentage of variables with complete information per outbreak, while crude variable completeness (*C_j_*) reflected the percentage of outbreaks with complete information per variable, such that:(1)$$C_{i}=\frac{j_{i}}{J}*100\%$$
(2)$$C_{j}=\frac{i_{j}}{I}*100\%$$
where *C_i_* and *C_j_*—completeness of *i-*outbreak or *j-*variable; *j_i_*—the sum of variables reporting complete information for *i*-outbreak; *J*—the total number of variables for which information was collected; *i_j_*—the sum of outbreaks reporting complete information for *j*-variable; and *I*—the total number of outbreaks for which information was collected.

### 2.4. Measuring Temporal Changes in Completeness

To examine trends in variable completeness over time, we sorted all 103 variables in descending order by crude completeness and grouped variables into 5 near-equal-sized categories (Appendix A). We defined category boundaries as follows: Category 1 (100–95% completeness; 18 variables), Category 2 (94–70% completeness; 21 variables), Category 3 (69–35% completeness; 22 variables), Category 4 (34–25% completeness; 20 variables), and Category 5 (24–0% completeness, 22 variables). 

Next, we calculated annual and monthly completeness for all outbreaks and for selected pathogens using monthly and yearly time series of outbreak counts and completeness values. We selected only 6 pathogen subgroups of interest: the 3 most reported pathogens in the study period (*Norovirus, Salmonella*, and *Clostridium*), 2 unique groups (outbreaks associated with unknown and multiple etiologies), and all pathogens (total reported outbreaks). We used yearly time series to examine trends in completeness across study years while monthly time series described the seasonality of completeness by Gregorian calendar month. We created yearly and monthly time series using the date of first reported outbreak illness. We calculated variable-based completeness for all outbreaks $N_{t}\;$and each selected pathogen ($N_{t,p}$), as: (3)$$C_{i,t}=\frac{r_{i,t}}{N_{t}}*100\%$$
(4)$$C_{i,t,p}=\frac{r_{i,t,p}}{N_{t,p}}*100\%$$
where $C_{i,t}$ and $C_{i,t,p}$—completeness for *i*-variable in *t*-time unit (*t* = 1–22 for annual completeness, *t* = 1–264 for seasonal completeness) for *p*-pathogens; $r_{i,t}\;$and $r_{i,t,p}$—outbreak records with complete information for *i*-variable in *t*-time unit for *p*-pathogen; $N_{t}$ and $N_{t,p}$—total count of outbreaks reported in *t*-time unit for *p*-pathogen. 

Once creating a time series of monthly completeness, we examined patterns of completeness over time, by a reported pathogen, and according to completeness categories, by estimating average completeness as:(5)$$A_{g,t,p}=\frac{S_{g,t,p}}{L_{g}}$$
where $A_{g,t,p}$—average completeness of *g*-category in *t*-month for *p*-pathogen; $S_{g,t,p}$—sum of monthly completeness of *g*-category in *t*-month for *p*-pathogen; and $L_{g}$—total number of variables included in *g*-category (ranging from 18 to 22).

### 2.5. Temporal Trend Analyses

To best capture trend differences between eFORS and NORS, we defined the segmented regression analysis critical point as January 2009 (onset of NORS reporting). We used segmented negative binomial regression models adjusted for linear monthly trends to estimate counts of outbreaks under both surveillance systems (Equation (6)):(6)$$\ln\left(E\left[N_{t,p}\right]\right)=\beta_{0}+\beta_{b}t_{b}+\beta_{a}t_{a}\;$$
where $N_{t,p}$—monthly number of outbreaks at *t*-month for *p*-pathogen; exponential of $\beta_{0}$—outbreak counts at the critical point (January 2009); $\beta_{b\;}\;\mathrm{and}\;\beta_{a}$—linear trend estimates for periods before and after the critical point, respectively; and *t_b_* and *t_a_*—continuous time series of study months from onset to the critical point and critical point to conclusion, respectively.

To examine trends in average monthly completeness, we used a segmented linear regression model adjusted for linear monthly trends and outbreak counts (Equation (7)) as well as harmonic regression terms (Equation (8)): (7)$$A_{g,t,p}=\beta_{0}+\beta_{1}t_{b}+\beta_{2}t_{a}+\beta_{3}N_{t,p}$$
(8)$$A_{g,t,p}=\beta_{0}+\beta_{b}t_{b}+\beta_{c,b}\cos\left(2\pi \omega t_{b}\right)+\beta_{s,b}\sin\left(2\pi \omega t_{b}\right)\;+\beta_{a}t_{a}+\beta_{c,a}\cos\left(2\pi \omega t_{a}\right)+\beta_{s,a}\sin\left(2\pi \omega t_{a}\right)$$
where$\;A_{g,t,p}$—average completeness of *g*-category in *t*-month for *p*-pathogen; $\beta_{0}$—average completeness at the critical point (January 2009); $\beta_{b\;}$and $\beta_{a\;}$—estimates of linear trends in completeness before and after the critical point, respectively; *t_b_* and *t_a_*—continuous study time series before and after January 2009, respectively;$\;\beta_{c}$ and $\beta_{s}$—harmonic trend coefficients for each critical period such that ω=1/M, where M is the length of the annual cycle in Gregorian calendar months (12). We determined seasonality by the presence of a significant sinusoidal or co-sinusoidal regression term.

We used Akaike Information Criterion (AIC) to examine model fit in Equation (6) and R^2^ values to examine model fit in Equations (7) and (8). We defined statistical significance in all analyses as α < 0.05. We performed data extraction, alignment, management, and cleaning using Excel 2016 Version 2103, Stata SE/16.1, and R (1.2.5033) software. We conducted statistical analyses using R (1.2.5033) software and created visualizations using R (1.2.5033) and Adobe Illustrator (25.4.1) software.

## 3. Results

### 3.1. Outbreak Frequency and Completeness by Pathogen

The 22,792 outbreaks in the study period were attributed to 41 contaminant groups: bacteria (*n* = 14), chemicals/toxins (*n* = 12), parasites (*n* = 7), viruses (*n* = 6), unknown etiology (*n* = 1), multiple etiologies (*n* = 1) (Table 1). Bacterial and viral outbreaks each accounted for ~30% of all outbreaks, followed by chemicals/toxins (~5%), and parasitic outbreaks (~1%). The most common pathogens were *Norovirus* (*n* = 6416, ~28%), *Salmonella* (*n* = 2872, ~13%), and *Clostridium* (*n* = 944, ~4%). Yet, most outbreaks reported unknown genus (*n* = 7401, ~32%) while the 5th ranked contaminant group was outbreaks with multiple genera (*n* = 779, ~3%). 

We found that NORS reported 88–103 variables per outbreak with variable-based crude completeness ranging 3.05–100.00% (Appendix A). Average crude completeness per outbreak varied from 41.96% to 85.37% for all contaminants (Table 1). Among pathogen groups, estimates were high for parasitic pathogens for both crude completeness per outbreak and per variable (66.05% and 64.10%, respectively). In contrast, outbreaks of unknown etiology had the lowest variable-based and outbreak-based crude completeness (51.22% and 46.06%, respectively). Within each pathogen group, variable-based completeness varied greatly: from 72.49% for *Cyclospora* to nearly half of this value at 38.86% for the other-parasite group. We found similar variation within the bacterial group, with 73.03% crude variable completeness for *Enterococcus* and only 48.23% completeness for *Shigella*.

We found distinct behaviors across completeness categories when examining outbreak completeness by pathogen groups (Table 1). Category 1’s completeness had the narrowest range (94.77–100.00%) across variables compared to other categories (Category 2: 52.38–100.00%, Category 3: 27.92–88.75%, Category 4: 0.00–76.19%, Category 5: 0.00–56.25%). Within each pathogen group, completeness per category varied broadly, especially in Category 5. For example, within the bacterial group, *Brucella* received 99.07% and 78.57% in Category 1 and 2, but only 6.25% in Category 5 whereas *Escherichia* received 97.69%, 75.28%, and 32.07% in these three categories, respectively. All contaminants showed a steady declining in completeness from Category 1 to Category 5. Furthermore, contaminants with higher completeness in Category 5 tended to have high completeness in other categories (e.g., *Toxoplasma*, *Cyclospora,* and *Sapovirus*). 

Location-related variables and total case counts reached 100.00% completeness across pathogens with >95% completeness for epidemiologic information related to illness symptoms. School-, beef-, and egg-related information had much lower completeness (ranging 10.00–20.00%) despite only being asked for a subset of outbreaks. Category 5’s variable on the secondary mode of illness transmission had the lowest completeness (3.05%) followed by etiology serotype and variables related to specimen testing types (6.98–9.42%). 

### 3.2. Annual Completeness

We identified an increased annual trend in average completeness for 21–31 pathogens consistently reporting outbreak characteristics (Figure 2). We found the lowest annual completeness in 1999 (36.99%, averaged from 26 pathogens) and highest in 2017 (78.18%, averaged from 27 pathogens). Average annual completeness increased by ~13% between the transition from eFORS in 2008 (50.15%) to NORS in 2009 (62.95%) (Figure 2, top panel). 

We found steadily increasing annual completeness patterns among pathogens with high outbreak counts when comparing across pathogens (Figure 2, bottom panel). We found the highest completeness of cases across pathogens after 2009, while the lowest cases always appeared before the critical point. For example, the top three annual completeness estimates were for *Giardia* in 2017 (*n* = 1, 97.75%), *Giardia* in 2019 (*n* = 1, 96.49%), and Rotavirus in 2019 (*n* = 1, 95.50%) while the lowest three annual completeness estimates were *Cyclospora* in 1998 (*n* = 1, 17.24%), other-bacterium outbreaks in 1998 (*n* = 1, 20.69%), and paralytic shellfish poison in 1999 (*n* = 1, 24.52%). Although the average annual completeness increased when surveillance transitioned from eFORS to NORS, not all pathogens demonstrated an instantaneous upward shift in completeness. For example, *Shigella* had 41.04% completeness in 2008 compared to 67.42% in 2009 whereas *Norovirus* had 55.16% completeness in 2008 compared to 57.44% in 2009. 

We found as total outbreak counts of individual pathogens went down, the fluctuation of annual completeness went up. For example, *Streptococcus* outbreaks received 52.95% completeness in 2015 compared with 65.31% in 2017 and 87.64% in 2014. For single reported outbreak across the whole study period, Monosodium glutamate (MSG) received 55.43% average completeness and *Enterococcus* received 73.03% average completeness. Therefore, pathogens with insufficient outbreak counts were not adequate to perform systematic completeness trend study. 

### 3.3. Segemented and Seasonality Trend Analyses

We examined the temporal trend for the three most-reported individual pathogens across five variable categories using monthly completeness value. We also selected outbreaks with unknown etiology to understand whether the missing of etiology information will lower the completion of other variables and outbreaks with multiple etiologies to determine if multiple pathogen outbreaks will increase the completion of other variables.

For each category, the average monthly completeness for all outbreaks showed a similar general trend to those of other selected pathogens (Table 2, Figure 3). Category 1 had ~100% completeness while other categories showed an increasing trend, especially in Category 4 and 5. Category 5’s average monthly completeness increased most after the transition to NORS with values increasing from ~0.00% completeness before 2009 to as high as 60.82% thereafter. We also saw large increases in average monthly completeness in Category 4, with values increasing from 1.28–35.99% during eFORS to 16.77–72.59% during NORS. Nonetheless, the completeness in outbreaks with unknown etiology remained low in Category 4 and 5 even after reporting system transition. In addition to the completeness trend, we observed a decrease in total outbreaks, *Norovirus* outbreaks, and outbreaks of unknown etiology after the transition to NORS. With respect to outbreak trend, outbreak counts for all pathogens decreased throughout the eFORS study period but remained stable during NORS (Figure 3 and Appendix A). The percent of change in outbreak counts by periods differed for each pathogen (Table 2). 

During eFORS, *Norovirus* had a 0.048% yearly increase while *Clostridium*, outbreaks with unknown etiology, and outbreaks with multiple etiologies decreased by 0.048%, 0.096%, and 0.048% per year, respectively. During NORS, we found a decreased annual trend for both *Norovirus* and outbreaks with unknown etiologies whereas outbreaks with multiple etiologies increased by 0.072% per year. We found no significant trend for either *Salmonella* during eFORS or *Salmonella* and *Clostridium* during NORS. 

Category 1 maintained high completeness despite differing outbreak counts in eFORS and NORS (Appendix A). The effect of outbreak counts on completeness differed by pathogen. We observed a decreasing trend in completeness as outbreak counts increased for outbreaks with unknown etiology. Except for Category 1, completeness was relatively higher in NORS than in eFORS (Appendix A). The highest monthly outbreak counts occurred in eFORS for all pathogens, outbreak with unknown etiology, outbreak with multiple etiology, *Norovirus*, and *Clostridium*.

We found that the monthly completeness of Category 1 and 2 reached over 60% for all pathogen groups when transitioning from eFORS to NORS (Table 3). We also found outbreak counts have limited association with completeness, with significant associations in Category 1 and 2. Outbreak counts were negatively associated with the average completeness of outbreaks with unknown etiology yet positively associated with outbreaks of multiple etiology. Furthermore, we found the greatest improvement in variable completeness for Category 2 and 3 during eFORS and Category 4 and 5 during NORS. For example, the monthly completeness for all outbreaks increased faster in NORS than in eFORS for Category 4 (0.197% in eFORS vs. 0.328% in NORS) and Category 5 (0.031% in eFORS vs. 0.396% in NORS), while monthly completeness increased faster in eFORS for Category 2 (0.160% in eFORS vs. 0.036% in NORS) and Category 3 (0.304% in eFORS vs. 0.078% in NORS).

The results of harmonic regression models (Appendix A) showed no seasonality in completeness across all categories for all outbreaks. Yet, we detected seasonality for *Norovirus* in Category 1 and 4 and for outbreaks of unknown etiology in Category 4 (*p* ≤ 0.037). These seasonal patterns in completeness were detected only during the NORS study period. 

## 4. Discussion

In this study, we described and evaluated variable completeness by pathogens and pathogen groups over time. Our findings provide essential information on data availability and suitability imperative for modelers performing time series analyses. Temporal patterns of this completeness metric illustrate substantial improvements in foodborne outbreak surveillance reporting over time after integrating surveillance system reporting under NORS. Furthermore, the annual completeness showed a steady trend in increasing completeness that exceeded 60% after 2009. We also observed improvements in completeness across variables, especially for those that contain specific characteristics rarely reported at the beginning of the reported period. 

The examination of average completeness by variable category is useful in assisting researchers with variable extraction and planning data analysis when using national outbreak surveillance data. Our findings suggest that NORS data are well equipped to study outbreaks’ general characteristics, such as outbreak location, eaten and preparation location, symptoms, and hospitalization information, as those variables had >70% average completeness (Appendix A). However, investigation and reporting could be improved for variables related to etiology and food products. Pathogen factors, such as a long incubation period, latent symptom onset, and delayed diagnosis, could potentially complicate outbreak investigations and the identification of pathogen etiology and contaminated food products. Our findings of low completeness, especially for pathogen etiology and food product variables likely highlight these challenges. Future research should focus on studying completeness patterns by food-related variables. Moreover, we noticed there were outbreaks where the completeness for certain related variables varied substantially (e.g., when variables should have been collected or missed simultaneously in the same outbreak report). For example, NORS had high completeness for the incubation period time unit but relatively low completeness for the incubation time itself (≳80% vs. ≲ 70%, respectively; Appendix A). Similarly, NORS had high completeness for the total number of cases and total primary cases, but low completeness for total secondary cases (~100% vs. 20.88%, respectively; Appendix A). Further improvements and validations could be performed by triangulating various data sources, say surveillance and hospitalization records, allowing detection of detailed discrepancies [29]. 

By examining pathogen completeness over time, data users can identify pathogens and pathogen groups with less missingness. We found that *Vibrio*, Scombroid toxin, and *Clostridium* had the highest average annual completeness. We also found that as the total outbreak counts of individual pathogen went down, the fluctuation of the corresponding annual completeness went up. This fluctuation is caused by the insufficient outbreak size. Therefore, we used the five most reported outbreak types for trend analysis. Among the five most reported outbreak types (outbreak of unknown etiology, *Norovirus*, *Salmonella, Clostridium*, and outbreak of multiple etiologies), *Clostridium* had the highest completeness followed by multiple etiologies, *Norovirus*, *Salmonella*, and unknown etiologies. It is very likely that data completeness is influenced by disease dynamics and diagnostic modalities implemented in the investigation protocols. Unfortunately, the metadata for both systems has very limited information on reporting capacity or testing rigor from the state and local facilities investigating each outbreak. No variables in eFORS and NORS describe the quality of reported records. For outbreaks of unknown etiology, the missing of etiological-related information will lower its average completeness. This could explain why the outbreak counts for unknown etiology decreased over time as the system improving. Moreover, after transition to NORS, data cleaning is more rigorous and NORS is more likely to distinguish the different modes of transmission (e.g., person to person, waterborne, foodborne, etc.) [30]. For example, a portion of *Norovirus* outbreaks, previously reported as foodborne, now is classified as non-foodborne outbreak [30]. In addition, we noticed an increasing trend in outbreak counts for outbreaks with multiple etiologies. Prior studies have indicated that outbreaks of multiple etiologies were more likely caused by the improper handle or environmental cross-contamination in cultured farms and dairy beef farms [31,32]. However, the reason for increased multiple etiology outbreaks in recent years remains unknown. 

In prior research, we found variations in seasonal peak timing across pathogens using harmonic regression modeling [33,34,35]. Such fluctuations could occur for a variety of social and environmental reasons [36,37,38,39]. Studies investigating temporal variations or seasonal patterns of illness depend on sufficient sample size for regression analyses to ensure proper statistical power [40]. The ability to detect pathogen’s seasonality is influenced by data aggregation and completeness [35,40]. Thus, the proposed metrics of completeness could be used as a tool for planning statistical analysis and determining needed statistical power when investigating foodborne illness or outbreak seasonality with surveillance data. Furthermore, we detected seasonality in the completeness of records for *Norovirus* and outbreaks of unknown etiology during the NORS period. The capability of identifying completeness seasonality in NORS might be indicative to the maturity of the surveillance system.

Our study was subject to several limitations and some of the limitations are due to the constraints of the surveillance reporting system. First, foodborne outbreak reporting was based on local government reporting standards and regulations, which may contribute to state-wide differences in categorizing foodborne and waterborne outbreaks [30]. These differences in reporting may lead to missing information related to outbreak contamination sources. Incomplete records could potentially occur when optional variables were not included in local investigation. Local variations in reporting practices could also affect outbreak grouping. For instance, some states may regard a multi-location outbreak as one combined outbreak, while other states report as several distinct outbreaks [41]. Some states report outbreaks using the broad CDC definition (the number of cases ≥2 per outbreak) while other states only report notifiable outbreaks [42,43]. When reporting practices depend on an outbreak size, reporting small outbreaks could better identify the sources of sporadic illnesses and disease patterns [44]. The inconsistency among outbreak definitions across states might prevent early outbreak detection and forecasting [45]. Outbreak, as a disease measurement term, needs to be more clearly and uniformly defined to better capture disease characteristics and detect disease patterns [45]. Second, as NORS is a dynamic system, the public health agencies can submit new or revise previous reports after new information becomes available [46]. Accordingly, the completeness results could vary depending on the time of data requests. 

Besides limitations due to the surveillance system, there are also limitations subjecting to our study design. For example, eFORS and NORS have a different variable definition and structure: eFORS contained 6 age groups whereas NORS contained 8 age groups. To evaluate the completeness between two periods the age-related variable, we had to use the average completeness across all age groups and thus reduced data granularity. During our data cleaning process, we excluded variables that were related to contact information, optional comments, and clarification responses due to its low relevance to study objectives. We further collapsed variables that were related to multiple-choice questions into single responses. Due to the vast number of variables and differences in variable structures, we were unable to examine all variables as their original structure presented in multiple surveillance reports. Lastly, we studied completeness at the outbreak event level. We did not investigate completeness for outbreak case information specifically because of reporting practices for public data. As case-based reporting is time consuming and labor intensive, public health agencies must balance cost-effectiveness and reporting accuracy. 

For decades, the government has collected foodborne disease outbreak information to investigate the occurrence, prevent the outbreak, and reduce the severity of foodborne illnesses. The United States Public Health Service and Centers for Disease Control (CDC) have been collecting and publishing periodate reports since the 1990’s [47]. The launch of eFORS provided valuable outbreak information, which was further enhanced by NORS [42,47]. The CDC has been improving the surveillance system through multiple actions. In October 1999, the CDC simplified its outbreak reporting form [48]. As a result, we observed an increase in average annual completeness in 1999–2000. As the laboratory and epidemiology methods have been improving over time, the completeness and level of details should improve in outbreak data [47,49]. Yet, unknown etiology and missing information are still present in the current surveillance system, which can undermine statistical power [50]. A good surveillance system improves outbreak information, reduces medical costs, better informs policies, and improves public health accountability [51]. As a passive surveillance system, the number of outbreaks reported by NORS are likely underreported. While active surveillance can provide more accurate and timely information, this type of surveillance system is expensive to maintain. To curtail missingness in outbreak surveillance systems, health practitioners and data curators could: Create a Standard Operating Procedure (SOP) to identify must-have variables, variables that are related to one another, and less-relevant variables. This SOP can assist in the streamlining of data cleaning procedures to identify true missingness, zero values, and information that is not applicable for an outbreak. Moreover, SOP can be used as a guideline to create NORS checkpoints to avoid missing information between related variables.Consider removing variables with consistently low completeness or conduct thorough investigation into the obstacles preventing adequate reporting these variables.Publicly report documentation explaining reasons for incomplete data; NORS has a rigorous data cleaning process that includes 30+ checkpoints for foodborne outbreaks. Outbreak data are reported as missing until all issues are solved [52]. Although incomplete outbreak reports cannot provide all information, these checkpoints and their completion may still be useful for researchers to study.In accordance with the Population Health Surveillance Theory, perform periodic system audits to evaluate data reporting procedure and data quality at the local level [53]. In addition, these periodic system audits can be used as an assessment to evaluate both workforce resource and laboratory testing capacities. For any local agency with low audit scores, the CDC can provide training materials, or relocate necessary recourses.

## 5. Conclusions

Information on secondary mode of illness transmission and specimen testing types had the lowest completeness in the assessed public surveillance data, yet such information could be of value to better understand the contribution of food products to outbreak etiology. Understanding completeness is essential in estimating statistical power and identifying the effective length of disease surveillance time series to examine disease trends and characteristics at the population level. Our completeness analysis is the first attempt to examine the missingness in publicly available national outbreak surveillance systems. Future work can assess completeness by variables or outbreak types across locations to better improve the outbreak information at the local level within NORS. The continuous improvement of surveillance records enables researchers to better utilize surveillance data and to model diseases with greater reliability.

## Figures and Tables

**Figure 1 ijerph-19-02898-f001:**
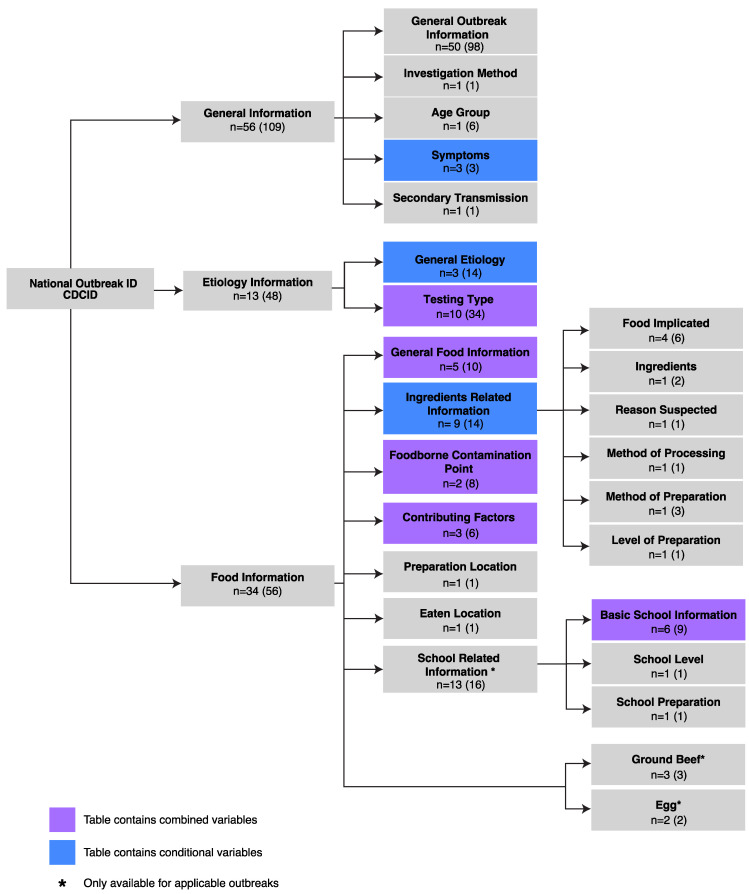
NORS foodborne outbreak data structure. CDCID is the unique identifier for each foodborne outbreak, characterized by 213 variables for 3 types of information and resulted in 103 variables after data cleaning and merging: general (*n* = 56 variables), etiological (*n* = 13 variables), and food-related (*n* = 34 variables). Variables were distributed over 13 sub-categories and data tables containing conditional (depicted in blue), combined (in purple), and unchanged variables (in gray). Numbers within the parentheses represent the original number of variables in each data table. Subcategories and data tables marked with an asterisk (*) indicate information only available for applicable outbreaks.

**Figure 2 ijerph-19-02898-f002:**
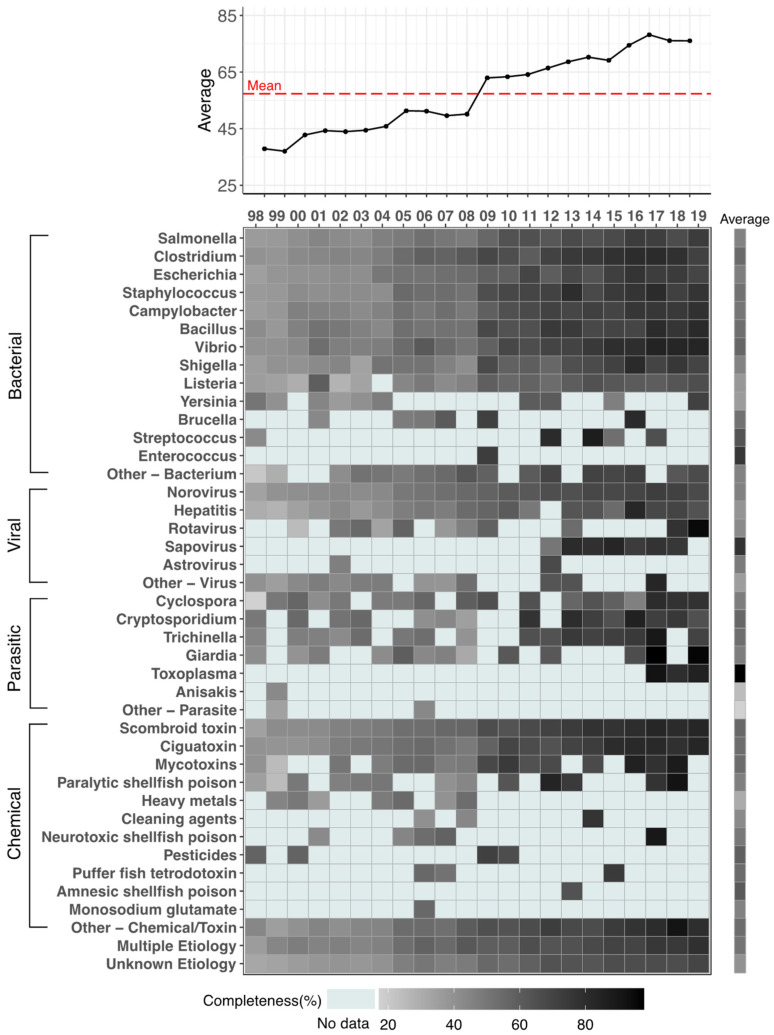
Average annual completeness across 103 variables by contaminant and contaminant type as reported by the National Outbreak Reporting System (NORS) in 1998–2019. The top panel provides a time series plot reporting the average annual completeness for all etiologies with the annualized mean indicated by the dashed red line. The bottom panel provides a heatmap with annual completeness estimates (**left**) and their average (**right**) for each of 41 contaminants. Blue color indicates outbreaks with no reported data for a given year while light grey and dark grey reflect low (~0%) and high (~100%) completeness, respectively. Contaminant group types are reported in descending order of outbreak counts.

**Figure 3 ijerph-19-02898-f003:**
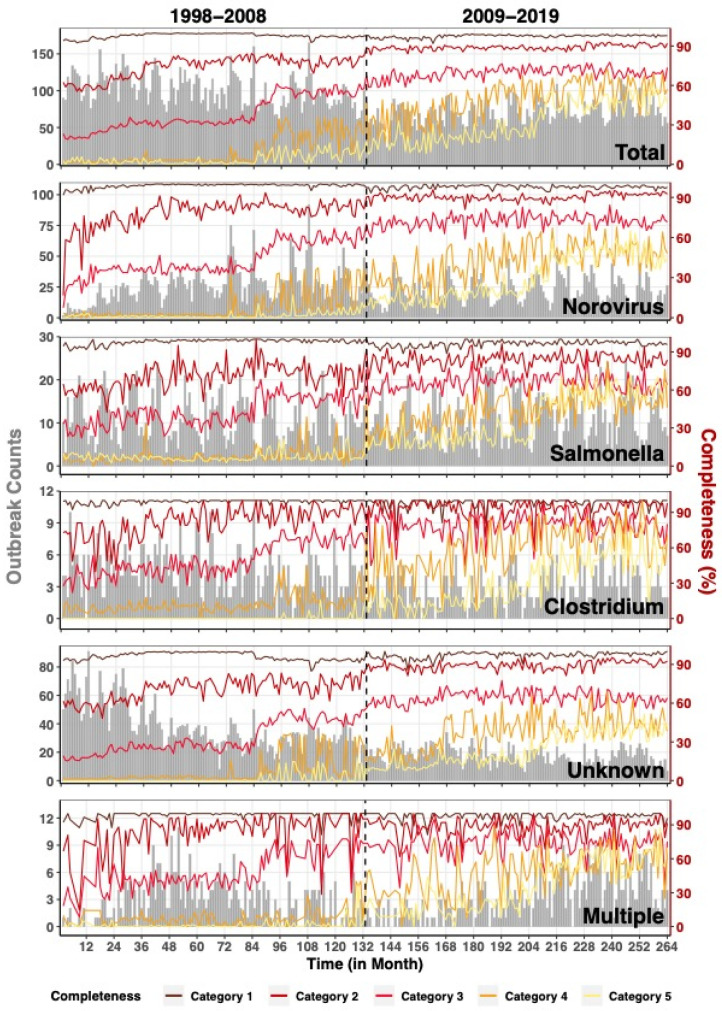
Shared-axis, multi-panel, stacked time series plots for monthly counts and average monthly completeness of foodborne outbreaks for all pathogens and by five contaminant subgroups as reported by the National Outbreak Reporting System (NORS) in 1998–2019. Each panel provides monthly counts of outbreaks (grey bars, left vertical axis) with superimposed time-series line plots reporting average monthly completeness (right vertical axis). We calculated monthly completeness as the average completeness of all outbreaks per month (as defined by illness onset date) for each completeness category (represented by colored lines from yellow (least complete variables) to red (most complete variables)). We report study months from January 1998 (1) through December 2019 (264).

**Table 1 ijerph-19-02898-t001:** Frequencies of outbreaks and completeness estimates for 41 contaminant groups publicly reporting data to the National Outbreak Reporting System (NORS) in 1998–2019. Completeness categories group variables by average crude completeness and are defined as: Category 1 (100–95% completeness; 18 variables), Category 2 (94–70% completeness; 21 variables), Category 3 (69–35% completeness; 22 variables), Category 4 (34–25% completeness; 20 variables), and Category 5 (24–0% completeness, 22 variables).

Contaminant Name	Number of Outbreaks	Crude Completeness		Completeness Category				
		per Outbreak	per Variable	1	2	3	4	5
All Outbreaks	22,792	59.45	53.65	98.45	81.00	52.48	29.01	14.46
Multiple Etiologies	779	64.54	60.19	98.61	87.72	60.09	34.91	25.56
Unknown Etiologies	7401	51.22	46.06	96.42	75.19	40.60	18.46	7.60
**Bacterial Pathogens**								
*Salmonella*	2872	58.94	53.90	97.16	77.20	52.12	27.08	22.41
*Clostridium*	944	62.37	56.90	98.83	85.91	59.62	32.99	13.91
*Escherichia*	649	58.69	56.18	97.69	75.28	51.29	30.69	32.07
*Staphylococcus*	625	56.02	49.69	98.84	78.39	48.04	21.57	9.29
*Campylobacter*	510	62.93	58.76	98.25	83.53	60.44	39.52	18.61
*Bacillus*	376	59.69	53.95	98.30	82.50	55.27	31.13	9.84
*Vibrio*	220	67.38	66.18	98.09	84.57	64.32	44.25	27.67
*Shigella*	195	53.61	48.23	97.08	74.77	46.32	18.78	7.92
*Listeria*	95	57.63	52.45	97.49	68.12	44.18	34.81	17.79
*Yersinia*	17	50.83	49.33	97.15	69.75	38.69	17.23	10.11
*Brucella*	6	61.39	59.13	99.07	78.57	63.96	32.14	6.25
*Streptococcus*	5	65.87	65.90	100.00	79.05	69.25	45.71	23.75
*Enterococcus*	1	74.42	73.03	100.00	100.00	85.00	57.14	6.25
Other—Bacterium	138	57.87	54.66	98.00	88.34	51.53	27.83	10.55
Subtotal	6653	59.55	54.59	97.81	79.40	53.59	29.31	19.54
**Viral Pathogens**								
*Norovirus*	6416	60.36	54.54	98.52	85.03	56.77	25.19	13.90
Hepatitis	103	47.08	44.13	96.55	62.23	34.07	19.73	8.00
Rotavirus	15	55.54	52.22	97.01	76.83	48.11	23.67	13.33
*Sapovirus*	15	73.41	66.79	96.30	91.75	67.84	33.33	42.75
Astrovirus	2	58.43	57.02	100.00	88.10	51.25	17.86	9.38
Other—Virus	102	45.59	38.64	97.26	67.65	29.86	2.84	0.88
Subtotal	6653	59.95	54.11	98.46	84.41	56.01	24.77	13.68
**Parasitic Pathogens**								
*Cyclospora*	112	70.58	72.49	98.54	85.63	67.79	62.10	46.95
*Cryptosporidium*	32	61.59	60.36	96.30	78.13	59.90	32.59	21.48
*Trichinella*	23	62.58	61.01	97.54	84.89	55.04	41.30	13.32
*Giardia*	22	53.63	48.42	98.94	71.65	45.70	13.86	10.43
*Toxoplasma*	3	85.37	85.86	100.00	100.00	88.75	76.19	56.25
Anisakis	1	44.57	43.07	100.00	57.14	36.67	7.14	0.00
Other—Parasite	2	41.96	38.86	100.00	52.38	27.92	0.00	0.00
Subtotal	195	66.05	64.10	98.15	82.47	61.69	43.00	33.32
**Chemicals and Toxins**								
Scombroid toxin/ Histamine	505	58.51	55.08	98.31	83.88	48.66	29.48	12.13
Ciguatoxin	349	61.97	59.12	98.74	84.84	49.61	41.98	13.39
Mycotoxins	35	62.78	57.25	98.24	88.03	58.27	26.57	10.59
Paralytic shellfish poison	17	56.43	54.45	94.77	80.95	44.94	25.63	11.40
Heavy metals	9	51.10	44.35	100.00	79.37	35.94	4.44	0.00
Cleaning agents	8	57.72	55.35	100.00	70.24	54.43	33.93	5.47
Neurotoxic shellfish poison	7	57.44	54.86	100.00	82.31	46.99	17.35	10.71
Pesticides	4	64.24	63.20	100.00	95.24	70.00	28.57	1.56
Puffer fish tetrodotoxin	3	62.65	60.54	99.79	92.06	57.92	33.33	2.08
Amnesic shellfish poison	1	66.57	64.33	100.00	95.24	51.25	57.14	6.25
Monosodium glutamate (MSG)	1	57.36	55.43	100.00	95.24	56.67	0.00	0.00
Other—Chemical/Toxin	172	56.37	50.73	98.65	83.19	47.95	23.76	7.83
Subtotal	1111	59.34	53.09	98.49	84.10	48.81	30.38	11.26

Bacterial, and certain viral and parasitic genes are italicized due to scientific nomenclature.

**Table 2 ijerph-19-02898-t002:** Monthly outbreak counts estimation by pathogen types (related to fitted curves in Appendix A). Results include the number of outbreaks at the time of system change, which is January 2009, and the yearly percentage change in eFORS and NORS study periods (with 95% confidence interval). LCI is the lower bound of the 95% confidence interval and UCI is the upper bound of the 95% confidence interval.

	Yearly % Change (eFORS)			Monthly Outbreaks Jan’09			Yearly % Change (NORS)		
Group	Estimate	LCI	UCI	Estimate	LCI	UCI	Estimate	LCI	UCI
All pathogens	−0.048 **	−0.060	−0.036	75.25	71.27	79.45	−0.0024	−0.012	0.012
*Norovirus*	0.048 **	0.024	0.060	30.01	26.47	34.01	−0.036 **	−0.06	−0.012
*Salmonella*	−0.012	−0.036	0.000	9.96	8.82	11.24	0.012	0.000	0.036
*Clostridium*	−0.048 **	−0.072	−0.024	3.18	2.76	3.64	−0.0036	−0.024	0.024
Unknown Etiology	−0.096 **	−0.108	−0.084	21.92	20.36	23.61	−0.036 **	−0.048	−0.024
Multiple Etiology †	−0.048 **	−0.072	−0.012	2.47	2.07	2.95	0.072 **	0.036	0.096

Estimation with ** represents *p*-value < 0.001. † Multiple etiology represents outbreaks with two or more confirmed or suspected etiology. Bacterial and certain viral genes are italicized due to scientific nomenclature.

**Table 3 ijerph-19-02898-t003:** Estimated monthly change in completeness effect by outbreak counts and system type for each pathogen groups and Category (related to Appendix A).

Category	Estimated % Completeness at the Point of System Changing		Estimated Effect Associated with Outbreak Counts		Estimated % Completeness Change in eFORS Time		Estimated % Completeness Change in NORS Time	
	Estimate	Std. Error	Estimate	Std. Error	Estimate	Std.Error	Estimate	Std. Error
**All Pathogens**								
1	97.943 **	0.161	−0.017 *	0.005	−0.002	0.002	0.001	0.002
2	87.826 **	0.489	−0.080 **	0.016	0.160 **	0.006	0.036 **	0.006
3	67.491 **	0.531	−0.228 **	0.017	0.304 **	0.007	0.078 **	0.007
4	26.317 **	0.781	−0.142 **	0.026	0.197 **	0.01	0.328 **	0.01
5	5.459 **	0.499	−0.038	0.016	0.031 **	0.006	0.396 **	0.006
** *Norovirus* **								
1	97.913 **	0.276	0.033 **	0.007	0.005	0.003	−0.009 **	0.003
2	87.420 **	1.13	0.052	0.029	0.155 **	0.011	0.025 *	0.011
3	67.965 **	1.098	−0.058 *	0.028	0.339 **	0.011	0.072 **	0.011
4	22.537 **	1.9	0.039	0.049	0.219 **	0.019	0.273 **	0.019
5	3.493 *	1.114	0.028	0.029	0.034 *	0.011	0.383 **	0.011
** *Salmonella* **								
1	97.946 **	0.388	−0.035	0.027	0.003	0.004	−0.007	0.004
2	80.611 **	1.229	−0.014	0.084	0.138 **	0.013	0.045 **	0.013
3	61.927 **	1.263	−0.139	0.086	0.254 **	0.014	0.068 **	0.014
4	21.740 **	1.609	−0.188	0.11	0.150 **	0.017	0.331 **	0.018
5	9.286 **	1.253	−0.071	0.086	0.021	0.014	0.432 **	0.014
** *Clostridium* **								
1	97.777 **	0.537	0.231 *	0.111	0.002	0.006	0.001	0.006
2	95.423 **	1.413	−0.245	0.292	0.190 **	0.017	−0.011	0.016
3	73.487 **	1.536	0.057	0.317	0.347 **	0.018	0.066 **	0.017
4	31.267 **	2.325	−0.584	0.48	0.212 **	0.027	0.417 **	0.026
5	0.335	1.465	0.479	0.303	0.028	0.017	0.445 **	0.016
**Unknown Etiology**								
1	96.679 **	0.45	−0.092 **	0.017	−0.048 **	0.006	0.029 **	0.005
2	85.505 **	0.919	−0.088 *	0.034	0.160 **	0.013	0.068 **	0.009
3	55.788 **	1.119	0.076	0.042	0.368 **	0.016	0.085 **	0.011
4	18.537 **	1.82	0.12	0.068	0.225 **	0.026	0.249 **	0.018
5	3.826 **	0.914	0.044	0.034	0.060 **	0.013	0.297 **	0.009
**Multiple Etiology**								
1	97.870 **	0.501	0.270 *	0.101	0.002	0.007	−0.012	0.006
2	88.028 **	1.940	1.189 *	0.391	0.138 **	0.026	−0.041	0.025
3	71.680 **	1.775	−0.580	0.358	0.335 **	0.024	0.049 *	0.023
4	22.670 **	2.225	−0.776	0.448	0.168 **	0.030	0.380 **	0.028
5	1.809	1.459	0.222	0.294	0.022	0.020	0.496 **	0.019

Estimation with ** represents *p*-value <0.001 and * represents *p*-value < 0.05. Bacterial and certain viral genes are italicized due to scientific nomenclature.

## Data Availability

We used publicly available datasets in this study, which are referenced throughout our manuscript text [13,14,15,16].

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
