# Peer review of "Evaluating Completeness of Foodborne Outbreak Reporting in the United States, 1998–2019"

_ijerph, 2022, doi:10.3390/ijerph19052898_

Round 1

Reviewer 1 Report

Overall it is a very interesting study. This manuscript indicated the first evaluation completeness of foodborne disease outbreaks in US from 1998 to 2019. This study is important for promoting to improve NORS.

I would recommend it for acceptance after following points are addressed on the manuscript.

  1. All outbreaks data were provided by CDC, but no CDC personnel was included in the co-authors.
  2. The English writing is not the best for native speaker. I suggest the authors go through the manuscript and streamline some sentences, especially the introduction part.
  3. Other comments

Line 38 approximately 1 in 6, not 1 and 6

Line 39 delete “~” in the manuscript

Line 40-41 add references. And Norovirus, Salmonella, Clostridium perfringens, Campylobacter spp. And Staphylococcus aureus are the top five pathogens contributing to foodborne illnesses in US.

Line 88 incubation periods, settings, or food sources.

Line 390-399 According to the findings, I suggest to add some advices and suggestions on improvement NORS through information application.

Line 465-478 add suggestions on improving capability of laboratory test and epidemiological investigation, in order to increase completeness

Author Response

Reviewer 1 Comments

Overall, it is a very interesting study. This manuscript indicated the first evaluation completeness of foodborne disease outbreaks in US from 1998 to 2019. This study is important for promoting to improve NORS. I would recommend it for acceptance after following points are addressed on the manuscript.

We thank the Reviewer.

All outbreaks’ data were provided by CDC, but no CDC personnel was included in the co-authors.

Thank you for the comment. While no authors are CDC personnel, we have provided an acknowledgement for the CDC personnel assisting us in data extraction as follows (Line 541-543):

“The authors would like to acknowledge the help of Ziming Dou for her support in data analyses and Hannah Lawinger, NORS Data Request Manager, for processing our request for extracted data.”

Additionally, we emphasize the importance of NORS as a publicly available data source throughout the text (Line 105, 108, 113, 117-119).

The English writing is not the best for native speaker. I suggest the authors go through the manuscript and streamline some sentences, especially the introduction part.

Thank you. We have edited the text with the help of native speakers, who are also co-authors of this paper.

Line 38 approximately 1 in 6, not 1 and 6

Thank you. We have revised.

Line 39 delete “~” in the manuscript

Thank you. We have kept the sign indicating approximations, as many of these estimates were reported with this disclaimer within reference [1-6].

Line 40-41 add references. And NorovirusSalmonellaClostridium perfringensCampylobacter spp. And Staphylococcus aureus are the top five pathogens contributing to foodborne illnesses in US.

We have added the reference [6].

Line 88 incubation periods, settings, or food sources.

Thank you. We have added “incubation periods” to the list (Line 89).

Line 390-399 According to the findings, I suggest adding some advice and suggestions on improvement NORS through information application.

We have revised the first recommendation in the Discussion section to emphasize the relevant suggestion as follows (Line 487-489):

“Moreover, SOP can be used as a guideline to create NORS checkpoints to avoid missing information between related variables.”

Line 465-478 add suggestions on improving capability of laboratory test and epidemiological investigation to increase completeness

We have revised the fourth recommendation in the Discussion section to emphasize the need for improving the capability of laboratory testing and epidemiological investigations as follows (Line 499-502):

In addition, these periodic system audits can be used as an assessment to evaluate both workforce resource and laboratory testing capacities. For any local agency with low audit scores, the CDC can provide training materials, or relocate necessary recourses.

Reviewer 2 Report

Overall comments:

The authors present an interesting structured manuscript examining the wholeness of data collected from outbreak surveillance systems. They show that a significant amount of relevant data are missing from these records. The authors provide suggestions on how this can be improved in the future, highlighting that the use of NORS has somewhat improved outbreak surveillance data completeness. Improvements in completeness of such data will ensure better predictive modelling and might improve outbreak prevention, detection and management.

Potential limitations:

Methods are not fully described making reproducibility of the study difficult. The study focused on highlighting that the data were incomplete but did not highlight exactly what data are missing and how the availability of this specific information would be useful in outbreak management. The authors did not highlight potential limitations of the NORS system that might be contributing to data incompleteness.

Comments and Suggestions for Authors

Abstract:

Line 15: Please define "US".

Line 24: What do the authors mean by disease serotype? specimen etiological testing?

Line 24-25: please explain why the variables with low (<35%) average monthly completeness during eFORS increased by 0.33-0.40%/month after transitioning to NORS.

Introduction:

Line 43 and Line: What does (USD 2013) and (USD 2018) mean?

Methods:

Some details on how analysis was done are missing hence it will be difficult to reproduce this study.

Line 116-135: NORS is mostly covered here, what about eFORS?

Line 119 to 120: what data were categorised as general? It is not clear how data were categorised; the authors need to clearly define how this was done.

How did the authors compare data collected by the two different systems? E.g., line 126: age groups? Line 181 to 185: what are the limitations of this approach? Please include these drawbacks under limitations in the discussion section.

Line 151-166: All variables excluded or collapsed during the cleaning process must be provided at least in the supplementary material to allow reproducibility of this analysis.

Line 165: Please provide more detail on these 103 variables to allow reproducibility of this analysis.

Results:

Table 1 and line 271-278: how would differences in outbreak numbers due to different pathogens e.g., Salmonella (n=2872) vs Enterococcus (n=1) or Bacteria (n=6653) vs Parasites (n=195) outbreak affect interpretation of completeness based on pathogen? Please comment on this in the discussion.

Line 287-289: how much of this is influenced by pathogen disease dynamics and diagnostic modalities?

Please highlight which data is missing the most from your analyses.

Discussion

Please comment: For some of the incomplete data e.g., lack of reported contaminated food sources, could it be incompleteness of data due to reporting insufficiency or due to disease or etiology or pathogen factors e.g., for Listeria monocytogenes which tends to have long incubation periods after consumption of contaminated food in some cases, making identification of outbreak associated food product difficult or impossible.

Line 400-407: how much of this is influenced by pathogen disease dynamics and diagnostic modalities?

Please discuss the potential limitations of the NORS system that might be contributing to data incompleteness.

Conclusion:

It is not clear which data is missing the most and how this data would be useful if available.

Author Response

Reviewer 2 Comments

The authors present an interesting, structured manuscript examining the wholeness of data collected from outbreak surveillance systems. They show that a significant amount of relevant data are missing from these records. The authors provide suggestions on how this can be improved in the future, highlighting that the use of NORS has somewhat improved outbreak surveillance data completeness. Improvements in completeness of such data will ensure better predictive modelling and might improve outbreak prevention, detection and management.

We thank the Reviewer.

Methods are not fully described making reproducibility of the study difficult. The study focused on highlighting that the data were incomplete but did not highlight exactly what data are missing and how the availability of this specific information would be useful in outbreak management. The authors did not highlight potential limitations of the NORS system that might be contributing to data incompleteness.

Abstract: Line 15: Please define "US".

We thank the Reviewer. We have spelled out this acronym at its first appearance.

Line 24: What do the authors mean by disease serotype? specimen etiological testing?

We have revised the text to avoid confusion as follows (Line 24):

“Variables with lowest crude completeness (ranging 3.32-6.98%) included pathogen, specimen etiological testing, and secondary transmission traceback information.”

Line 24-25: please explain why the variables with low (<35%) average monthly completeness during eFORS increased by 0.33-0.40%/month after transitioning to NORS.

We have clarified the potential reasons for low completeness as follows (Line 27-28):

“Variables with low (<35%) average monthly completeness during eFORS increased by 0.33-0.40%/month after transitioning to NORS, most likely due to the expansion of surveillance capacity and coverage within the new reporting system.”

Introduction: Line 43 and Line: What does (USD 2013) and (USD 2018) mean?

The statement of USD 2013 and USD 2018 refers to American Dollars in 2013 and 2018 values. In other words, these estimates are not inflation adjusted.

Methods: Some details on how analysis was done are missing hence it will be difficult to reproduce this study.

Line 116-135: NORS is mostly covered here, what about eFORS?

Thank you for the comment. When NORS launched in 2009, eFORS data were integrated within the NORS reporting system, therefore data collected from 01 January 1998 through 31 December 2008 belong to the eFORS system. We have revised to clarify this transition as follows (Line 117-119):

“On 04 March 2021, we requested and received integrated NORS data for all available foodborne outbreak records from 01 January 1998 through 31 December 2019 [28], where records between 01 January to 31 December 2008 were collected by eFORS.

Line 119 to 120: what data were categorized as general? It is not clear how data were categorized; the authors need to clearly define how this was done.

Thank you for the comment. We adopted the categorization of variables as reported by eFORS and NORS. This includes subgrouping reported records into 3 types of information (general, etiological, and food-related), which correspond to specific variables and records tables within circulated data. We define these categories in Line 119-137. We also provided Reference 28 (Line 118), which is the original data source that has more detailed information about eFORS and NORS data structure.

To avoid confusion, we restated the sentence in question as follows (Line 126):

“NORS records related to general outbreak information (e.g., location, illness date, incubation time, and case information)…”

How did the authors compare data collected by the two different systems? E.g., line 126: age groups? Line 181 to 185: what are the limitations of this approach? Please include these drawbacks under limitations in the discussion section.

Thank you for this very valuable point. Upon receipt, we harmonized and aligned records from both systems to follow the same format. We compared the same variables in both eFORS and NORS reporting systems, yet some variables, like age, were presented in different ways. The two systems had differing definitions for age variables, with eFORS reporting 6 groups and NORS reporting 8 groups. To standardize completeness estimates across both systems, we calculated average completeness across all age groups, and thus we reduced the data’s granularity because we were unable to determine the improvement of completeness in individual groups. We have added this limitation in the Discussion section as follows (Line 459-462):

Third, eFORS and NORS have a different variable definition and structure. For example, eFORS contained 6 age groups whereas NORS contained 8 age groups. To evaluate the completeness between two periods the age-related variable, we had to use the average completeness across all age groups and thus reduced data granularity.”

Line 151-166: All variables excluded or collapsed during the cleaning process must be provided at least in the supplementary material to allow reproducibility of this analysis.

Thank you for this very valuable point. We have provided these variables and the cleaning process used in Supplementary Table S1.

Line 165: Please provide more detail on these 103 variables to allow reproducibility of this analysis.

Thank you for this very valuable point. We have provided these variables and the cleaning process used in Supplementary Table S1.

Results:

Table 1 and line 271-278: how would differences in outbreak numbers due to different pathogens e.g., Salmonella (n=2872) vs Enterococcus (n=1) or Bacteria (n=6653) vs Parasites (n=195) outbreak affect interpretation of completeness based on pathogen? Please comment on this in the discussion.

Thank you for your valuable comment. We found that as the total number of outbreaks for an individual pathogen increased the completeness declined in some specific cases. As shown by the regression analysis, “we also found outbreak counts have limited association with completeness, with significant associations in Category 1 and 2. Outbreak counts were negatively associated with the average completeness of outbreaks with unknown etiology yet positively associated with outbreaks of multiple etiology” (Line 360-364).

We commented on these findings in the Discussion section as follows (Line 411-414):

We also found that as the total outbreak counts of individual pathogen went down, the fluctuation of the corresponding annual completeness went up. This fluctuation is caused by the insufficient outbreak size. Therefore, we used the five most reported outbreak types for trend analysis.

Line 287-289: how much of this is influenced by pathogen disease dynamics and diagnostic modalities?

Excellent comment. Differences in completeness by category and pathogen groups may be driven by differences in diagnostic modalities across reporting agencies and disease dynamics. Unfortunately, the metadata for both systems have very limited information on reporting capacity or testing rigor from the state and local facilities investigating each outbreak. We have commented on this limitation in the Discussion section as follows (Line 417-421):

It is very likely that data completeness is influenced by disease dynamics and diagnostic modalities implemented in the investigation protocols. Unfortunately, the metadata for both systems have very limited information on reporting capacity or testing rigor from the state and local facilities investigating each outbreak. No variables in eFORS and NORS describe the quality of reported records.

Please highlight which data is missing the most from your analyses.

Thank you for your comment. We have described which variables have the most missing data in the Results section (Line 293-299) and the Discussion section (389-393). In addition, we added a sentence to the Conclusion section as follows (Line 504-506):

Information on secondary mode of illness transmission and specimen testing types had the lowest completeness in the assessed public surveillance data, yet such information could be of value to better understand the contribution of food products to outbreak etiology.

Discussion:

Please comment: For some of the incomplete data e.g., lack of reported contaminated food sources, could it be incompleteness of data due to reporting insufficiency or due to disease or etiology or pathogen factors e.g., for Listeria monocytogenes which tends to have long incubation periods after consumption of contaminated food in some cases, making identification of outbreak associated food product difficult or impossible.

We thank the Reviewer for this valuable point. We agree that long incubation periods might make traceback investigations harder and less feasible for identifying contaminated products. However, to conclude that long incubation periods can cause incomplete food information, future research will need to explore completeness patterns by food-related variables, not completeness categories. We have commented on this in the Discussion section as follows (Line 392-398):

“However, investigation and reporting could be improved for variables related to etiology and food products. Pathogen factors, such as a long incubation period, latent symptom onset, and delayed diagnosis, could potentially complicate outbreak investigations and the identification of pathogen etiology and contaminated food products. Our findings of low completeness especially for pathogen etiology and food product variables likely highlight these challenges. Future research should focus on studying completeness patterns by food-related variables.

Line 400-407: how much of this is influenced by pathogen disease dynamics and diagnostic modalities?

Thank you for your comment. As noted above, we are unable to determine how patterns of completeness over time are influenced by disease dynamics or diagnostic modalities. We have added this comment in the Discussion section as follows (Line 417-421):

It is very likely that data completeness is influenced by disease dynamics and diagnostic modalities implemented in the investigation protocols. Unfortunately, the metadata for both systems have very limited information on reporting capacity or testing rigor from the state and local facilities investigating each outbreak. No variables in eFORS and NORS describe the quality of reported records.

Please discuss the potential limitations of the NORS system that might be contributing to data incompleteness.

Thank you for your comment. We added a potential limitation of the NORS system that might be contributing to data incompleteness in the Discussion section as follows (Line 459-462):

Third, eFORS and NORS have a different variable definition and structure. For example, eFORS contained 6 age groups whereas NORS contained 8 age groups. To evaluate the completeness between two periods the age-related variable, we had to use the average completeness across all age groups and thus reduce data granularity.

Conclusion:

It is not clear which data is missing the most and how this data would be useful if available.

Thank you for your comment. We added the following sentence to the Conclusion section as follows (Line 504-506):

Information on secondary mode of illness transmission and specimen testing types had the lowest completeness in the assessed public surveillance data, yet such information could be of value to better understand the contribution of food products to outbreak etiology.”

Round 2

Reviewer 2 Report

The paper is much improved. However, the authors did not highlight potential limitations of the NORS system that might be contributing to data incompleteness. Please discuss these potential limitations.

Author Response

Response to Reviewer Comment

The paper is much improved. However, the authors did not highlight potential limitations of the NORS system that might be contributing to data incompleteness. Please discuss these potential limitations

Thank you for your suggestion. We have recognized that Reviewer is asking to highlight the deficiencies of NORS system that are contributing to limited completeness. Thus, in the revised version we separated the limitations of the NORS and limitations associated with the study design and execution. We expanded potential limitations of the NORS system in the Discussion section as follows (Line 445-453 and Line 472-482):

“Our study was subject to several limitations and some of the limitations are due to the constraints of the surveillance reporting system. First, foodborne outbreak reporting was based on local government reporting standards and regulations, which may contribute to state-wide differences in categorizing foodborne and waterborne outbreaks [30]. These differences in reporting may lead to missing information related to outbreak contamination sources. Incomplete records could potentially occur when optional variables were not included in local investigation. Local variations in reporting practices could also affect outbreak grouping. For instance, some states may regard a multi-location outbreak as one combined outbreak, while other states report as several distinct outbreaks [41].”

Besides limitations due to the surveillance system, there are also limitations subjecting to our study design. For example, eFORS and NORS have a different variable definition and structure: eFORS contained 6 age groups whereas NORS contained 8 age groups. To evaluate the completeness between two periods the age-related variable, we had to use the average completeness across all age groups and thus reduced data granularity. During our data cleaning process, we excluded variables that were related to contact information, optional comments, and clarification responses due to its low relevance to study objectives. We further collapsed variables that were related to multiple-choice questions into single responses. Due to the vast number of variables and differences in variable structures, we were unable to examine all variables as its original structure presented in multiple surveillance reports.”